# Tourist Motivations and Satisfaction in the Archaeological Ensemble of Madinat Al-Zahra

**Amalia Hidalgo-Fernández, Ricardo Hernández-Rojas** **, Juan Antonio Jimber del Río *** and **José Carlos Casas-Rosal**

Faculty of Economics, University of Cordoba, C.P. 14002, Plaza Nueva S/N, 14071 Córdoba, Spain; es1hifea@uco.es (A.H.-F.); et2heror@uco.es (R.H.-R.); jcasas@uco.es (J.C.C.-R.)

*   Correspondence: jjimber@uco.es; Tel.: +34-657-682-551

**Abstract:** Madinat Al-Zahra is an ancient Arab palace located in Cordoba, Spain, which was proclaimed as part of the World Heritage of Humanity in 2018. The public administration must know the tourist motivations and interest in this heritage, as well as the satisfaction with Cordoba as destination and this Archaeological Ensemble. This article addresses these issues by analyzing and comparing motivations, expected and perceived value with the destination and those of the monument itself, using a Structural Equation Models methodology. Data was obtained from a survey of 375 tourists and the results showed that there is a high satisfaction with both destination and site, although these people perceive that this heritage is not integrated with the rest of the city attractions. In addition, the survey highlights that the transport system to the site is one of the most important weaknesses, among others. The results of this study offer valuable information related to this heritage.

**Keywords:** heritage tourism; motivations; satisfaction; Madinat-Al Zahra; World Heritage Site

---

## 1. Introduction

Archaeological tourism is a specific type of cultural tourism, which enjoys great acceptance and relevance in the area and helps to raise awareness of this type of heritage importance; besides, it makes possible regional revitalization all around the world. Not in vain, in recent years, archeology has contributed to the strengthening of new urban economies supported by tourism.

As shown in [1], archeology is a fundamental basis to elaborate strategies focused on tourist development of cities and territories with an historical and cultural past.

The work of [2] reveals in addition how archeology assigns symbolic values to the tourist area because of a reflection on its role in any circumstance, whether tourist or not.

An archaeological site cannot become a tourist resource if it has not been the object of an intervention that guarantees its conservation and allows visitor understanding. Regarding the impact that this type of heritage has on its surroundings, in one study [3] the visited heritage facilities, the conservation, cleanliness, and security of the surrounding city are identified, as well as the most influential factors for the potential tourist interested in the archaeological heritage of an area.

Cordoba, Spain, is one of the eight Andalusian provinces; its historical and heritage wealth allows it to be one of the Spanish cities that have a consolidated demand for cultural tourism, as with Salamanca, Santiago de Compostela, Toledo, and most of the World Heritage cities [4]. All these cities receive a high number of visitors per year without promoting tourism.

Cultural tourism is a growing sector, not because of a great interest in culture but due to its historical and heritage value [5]. In Cordoba, there are numerous attractive heritage sites for cultural tourism, which is reflected in its cultural wealth and is represented by four inscriptions in the World Heritage Site by UNESCO. The first inscription was the Mosque-Cathedral in 1984. In 1994,

it was extended to the entire historic center of Cordoba (Old Jewish Quarter). The third was in 2012, when the Courtyard Festival was proclaimed as World Heritage of an immaterial nature and the fourth was granted to the Archaeological Ensemble of Madinat Al-Zahra in July 2018. Regarding this last inscription, [6] on the history of the Archaeological Ensemble of Madinat Al-Zahra from a historical and geographical approach, as well as researching tourism in Europe, reference [5] show us the importance of adopting general measures of action for the management and control of tourism that lead to an increase in economic activity.

The objective of this research is contributing to the knowledge of the Archaeological Ensemble of Madinat Al-Zahra from the tourist point of view, the motivations that lead to visiting it, and the overall satisfaction with regard to the Archaeological Ensemble of Madinat Al-Zahra, regarding its own destiny and with respect to the city of Córdoba.

After this introduction, we present the theoretical framework which includes a review of the literature in the field of World Heritage monuments from the tourist point of view, as well as the motivations and the satisfaction of the tourist with the destination and a brief history of the Caliphate's city. Afterwards, the research model and the methodology used are presented in order to show the results and discussion. Finally, the most relevant conclusions of the investigation are presented.

## 2. Theoretical Framework

### 2.1. Tourism and World Heritages Sites

Some of the world's well-known and consolidated heritages are the Archaeological Park of Petra (Jordan) [7], The Notre Dame Cathedral in Paris, The Cathedral of Santiago de Compostela, The Sagrada Familia in Barcelona [8], and The Blue Mountains of Australia [9].

In countries such as Italy [10], we observe that this recognition does not influence the increase in the number of tourists, although it does have an effect on the monument itself, increasing its visits and entering direct competition in the list of UNESCO's tourists. Since these sites have power enough to motivate movement and enjoyment, they can even help to bring visitors to other unlisted attractions, within their limits [11]. The increase in the number of tourists to the well-named World Heritage Sites occurs mainly from foreigners to the country [12].

In relation to the scientific texts about tourism, in which this activity is related to the heritage of humanity [13], they identify more than 900 articles in the period between 2008 and 2014. The authors analyzed a total of 178 contributions that address the general issue and case studies, for both consolidated and emerging destinations. They differentiate three large groups of publications. The first includes contributions that address general and conceptual aspects of tourism and World Heritage sites [14–16]. The second group consists of studies conducted in consolidated destinations [8,17,18], and a third one refers to the studies of destinations that are in consolidation [19–21] and that find an element of local development in tourism [22]. The investigations related to the Archaeological Ensemble of Madinat Al-Zahra would be part of the studies that were carried out in already established destinations. However, there are no specific studies of this group so this research is a unique and unprecedented contribution from the tourism point of view of the new World Heritage, granted in 2018.

Currently, Cordoba has four inscriptions considered World Heritages Sites by UNESCO. The first inscription was in 1984 with the designation of the Mosque-Cathedral. Ten years later, it was extended to the entire historic center (Old Jewish Quarter). These two first concessions had two effects: the tourist rediscovery of the historic city, with the strengthening and increase in cultural tourism, as has happened in other heritage cities [4]; and greater interest from the authorities towards the city's tourist and cultural potential, creating the need for management and sustainable development. The third inscription was obtained in 2012, when the Courtyard Festival was listed as World Heritage of an intangible nature. This festival has been studied since its origins and there has been research on its tourist evolution from the point of view of the socio-demographic profile, the perceptions of

foreign tourists that participate in this cultural event [23], and from the development of tourism based on an intangible heritage. Therefore, Cordoba is defined as a consolidated destination in terms of World Heritage.

## 2.2. Tourism Satisfaction

The final result and the experience of the tourist after visiting heritage of humanity sites have been studied from different perspectives. From the point of view of the motivation for the trip, it contributes to knowing why studying the motivations for a cultural visit is important [24]; likewise, culture is one of the main motivations, therefore, it is necessary to analyze both the destination and the tourist motivations to visit a place [16].

The visitor satisfaction is important due to the fact that it is one of the most important assessments for future visitor behavior, as shown by numerous studies [25,26]. Likewise, in the study on the key characteristics of the culture of a Welsh island and sustainable tourism in Wales, the authors [27] emphasize the importance of the correct management of the tourist or visitor, which affects the satisfaction of the tourist, in order to avoid an internal weakness of the site of the visit. In the same way, the authors [28] propose different indicators of tourism sustainability in the economic field, including tourist satisfaction.

The overall satisfaction along with the tourist expectations are related to the induced benefit in the cities inscribed in the world heritage list, because of their international tourist positioning [29] since they are the characteristics that justify their inclusion as Heritage (exceptionality, uniqueness, universality, and authenticity, among others) that make them a tourist attraction [30,31]. There are studies that relate satisfaction in tourism with heritage [32], which have revealed that the tourist satisfaction with the individual components of a destination leads to their overall satisfaction [33]. Consequently, this study considers the overall satisfaction of the tourist both with respect to the archaeological site of Madinat al-Zahra and Cordoba as destination.

Reference [34] shows in the results of their work that the modality of group visits to Madinat Al-Zahra has experienced a considerable loss in relation to its affluence, determining that new ways of structuring visits from a local perspective would be necessary, it would be also important to take into account that tourists stayed only for one hour on several occasions.

In the work carried out by [35] they investigated through two questionnaires for foreign and Spanish visitors: one for individual visitors (2471 surveys conducted) and another for group visitors (371 surveys conducted) on the guidelines of tourist cultural consumption of archaeological heritage. This is meant to provide keys to understand the characteristics of the visitors in these places, the tourist reading that makes use of these resources, and to understand—in the future—the real role played by this type of heritage in the tourist destinations they are part of.

The concept of perceived value has shifted towards an approach based on tourists, according to which "quality resides in the eyes of the recipient". Quality is what customers perceive it to be, they are the ones who observe and determine if a service is quality or not. According to this personal and subjective point of view of quality, many of the definitions that are currently used go around the idea that the perceived quality of a service is a global consumption judgment, in relation to superiority of services [36] that results from the comparison made by clients between their expectations about the service they will receive and the perceptions of the organization of those services.

The perceived quality is composed of opinions regarding the quality of the received services, the quality of the accommodation, the quality of catering, the quality of leisure and entertainment, and the quality of the experience in general.

The study model is based on the theoretical foundations of the American Customer Satisfaction Index (ACSI). The ACSI uses periodic studies to measure the domestic consumer satisfaction with American products and services. Satisfaction is measured on a scale of zero to 100, the resulting index is extremely useful for comparing companies in the same sector [37] and others [38]. The current model, however, has some new and different characteristics. A novelty of this model is the type of

variables to calculate the perceived satisfaction. Tourism expectations include the motivations and interest of travelers, since travelers have a significant influence on this latent variable.

The perceived value is the consumers' understanding of the product advantages [39]. According to [40], it is the evaluation of the clients about how the product responds to what was expected. Once the tourist goes on a trip, he/she has expectations that he/she expects to see fulfilled, which leads to a high level of satisfaction.

### 2.3. The Caliphate of Cordoba

The history of the Caliphate city, originally called Madinat al-Zahra, began in 711, when the Muslims of the Umayyad caliphate invaded the Iberian Peninsula, they were conquerors of the Visigothic kingdom that occupied almost the entire peninsula. Al-Andalus emerged in Spain as a consequence of this invasion, it was a new province of the Muslim empire or "Islamic caliphate." This territory was governed by an emir, in charge of spreading the propositions of the Muslim Caliph of the Umayyad dynasty. In the 8th century, this dynasty received aggression from the Abbas family, which took control of the Caliphate government. After the defeat, the Umayyads decided to settle in Al-Ándalus, specifically in Córdoba, its capital, and Abd-al-Rahman was proclaimed emir of this new independent emirate [41].

A few years later, the caliphate of Cordoba was created as a completely independent state from the rest of the Islamic empire. Abd al-Rahman III proclaimed himself as the first caliph of Cordoba, which entailed the religious and political power of the territory. He needed an army and to control the society to avoid revolts promoted by his censors. At first, his place of residence was the Fortress of Cordoba, located in the Mosque, but a few years later, in 936, he started building his own palace, which he named Madinat al-Zahra, which was his place of residence as well as that of all the caliphates that followed him. It was active for 80 years [42], until it was destroyed by Almoravids invaders.

To reveal the political, historical, social, economic, and cultural dimension of Madinat Al-Zahra, we should focus on the tenth century, in an Islamic world divided in three great rival caliphates at a political level, the Abbasid with its capital in Baghdad, the Fatimí with its headquarters in Cairo, and the Andalusian Umayyad with its capital in Cordoba, in which the Palace of Madinat al-Zahra showed its power and splendor.

This caliphal metropolis was adapted for its inhabitants with road construction, water supply, and everything necessary for the new city. Madinat al-Zahra consisted of three parts: the medina, where the merchants, artisans and peasants lived; the alcazar, where the rulers, the crown prince, the caliph, and the servants resided; and the third part that was of religious scope, the Aljama Mosque [43].

Therefore, based on its important historical past, which is shown in [44], we can affirm that the ancient Caliphate city of Madinat al-Zahra has an international cultural and tourist interest, which is supported and reflected with the inscription as a World Heritage Site by UNESCO, thus endorsing the cultural importance of the area.

### 3. Methodology

### 3.1. Destination

Spain is geographically located as the southernmost country of the European continent, it has 50 provinces and it is composed of 17 Autonomous Communities along with two autonomous cities (Ceuta and Melilla), it has 46.54 million inhabitants in total. Cordoba is located in the south of Spain and is well connected by train (Spanish high speed) and road. It is one of the eight provinces that make up the Autonomous Community of Andalusia. It currently has 788,219 inhabitants [45] and the tertiary sector is its main activity, highlighting tourism. The Archaeological Ensemble of Madinat Al-Zahra is located 10 km west of the capital at the foot of the Cordoba Hills. It has a Museum, a Visitor Reception Center and the archaeological site itself that is 800 m from there. The access to the Visitor Reception

Center, which was opened in 2009 along with an interpretation museum, is by one's own vehicle or by bus, from there to the Archaeological Ensemble of Madinat Al-Zahra there is an exclusive bus called a "shuttle".

In terms of tourism, more than eleven million foreign tourists visited Andalusia in 2017. This autonomous community is the fourth Spanish tourist destination, it is only surpassed by Catalonia (13 million tourists), and the Balearic Islands and Canary Islands (12 million) [46]. The transformation of Andalusia as a destination that integrates culture with the traditional sun and beach concepts is an already studied fact corroborated by several authors [47].

In the national ranking of tourists, Cordoba was in the 22nd position out of 50 provinces in 2017, with 1,232,004 tourists [45]. This is a seasonal demand, the month of May being the one of greater affluence and December the one of lesser importance. The proper management of patrimonial and cultural spaces can contribute to the population's social and economic development, as well as strategic proposals to enhance visits, optimizing management, which can revive the local economy.

As can be seen in Figure 1, the average percentage of tourists visiting the site, among all those who choose Cordoba as a destination, is 17.91%, with stable values at 15% since 2014. The percentage of tickets for Madinat-Al-Zahra is lower compared to the total sold for monuments of the city, with an average value of 7.55%, but located below 6% since 2015.

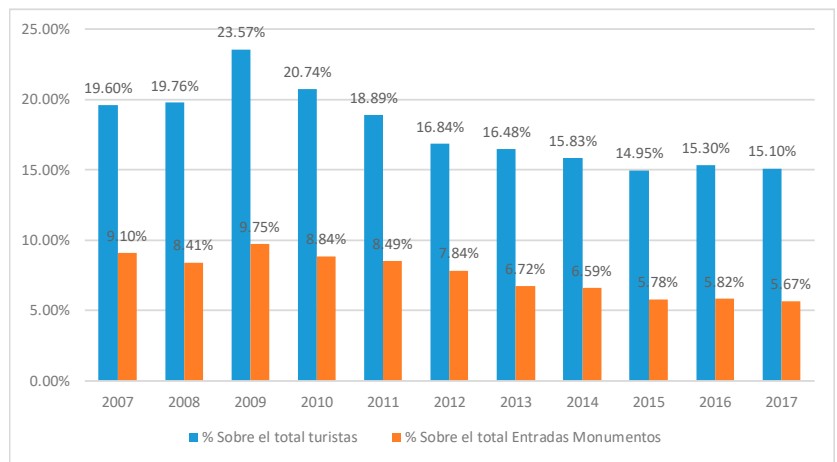

**Figure 1.** Percentage weight on the total of tourists in Cordoba and on the total of tickets for Monuments. Source: authors' elaboration based on the Tourism Consortium data. Instituto municipal de turismo de Córdoba.

The evolution of the number of tickets sold in the Architectural Ensemble of Madinat Al-Zahra in the last ten years is reflected in Figure 2, which also highlights the different events that may have intervened in the produced changes. The "Daily Bus to Madinat al-Zahra" service was launched before 2006 in order to ease the visit, and, according to the Cordoba observatory data, about 17000 tourists use this service annually. Subsequently, the online sales system was implemented in 2006 allowing people to book a place on this bus.

Between 2007 and 2010, the Tourist Observatory of Cordoba, in the annual report prepared on the study of the demand and the characteristics of the tourist model of the city, detected that tourists who visited Cordoba in that year were mostly Spanish, and there was a slight decrease in foreign tourism compared to 2006 as a result of new tourist circuits opening in Central Europe.

In 2010, the international economic crisis was reflected in Cordoba, therefore consumption and spending capacity were detracted. This means that, in 2009 and 2010, there was a decrease of 7.02% in the number of visits compared to previous years due to the economic crisis. When observing the demographic characteristics of tourists that visit Cordoba, it stands out that 61.94% are Spanish, 30.30% come from European Union countries, and 7.76% from the rest of the world. The Andalusian Community is the one that brings more national tourists (25.69%), although it decreases compared

to 2008, followed by the Community of Madrid (21.69%), which grows compared to 2008. It also highlights the increase in visits by tourists from Catalonia (9.61%), Valencia (9.22%), and Castilla León (8.82%). In reference to the country of origin of non-national tourists belonging to countries of the European Union, France (26.51%) is in the first place, although it decreases compared to 2008, the second place is for the Italian market (14. 86%), and the third place for the German one (12.05%).

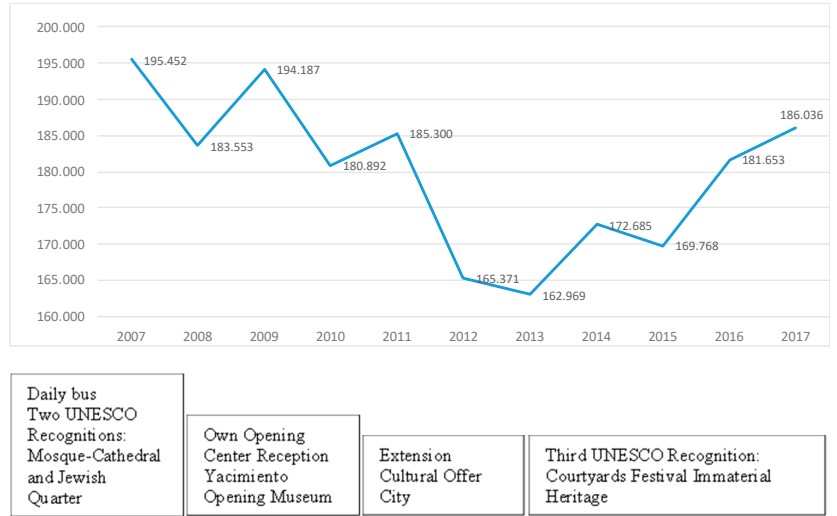

**Figure 2.** Tourist entries in the last ten years to the Archaeological Ensemble Madinat Al-Zahra plus more significant effects from the tourist point of view. Source: authors' elaboration based on data from the Tourism Consortium.

At the end of 2009, the Cordoba Convention Bureau Program was launched to promote Cordoba as a venue for congresses, conventions, incentive trips, meetings, and events, but although it initially attracts tourists, it will not have the expected effect in later years due to a lack of infrastructure.

In 2010, the cultural and patrimonial offer increased with the opening of the Mudejar Chapel of the Church of San Bartolomé, located in the Faculty of Philosophy and Letters. The project "The soul of Cordoba" began, with night visits to Cordoba's Cathedral, the old Mosque, the permanent equestrian show of the Royal Stables "Passion and Elf of the Andalusian Horse", and the "Gastronomic Cordoba" program, whose objective is to promote the city as a quality gastronomic destination.

In 2011, there was an increase in visits to Madinat al-Zahra. The projects of the previous year were consolidated and so began "Cordoba, the light of cultures", a night-time show of water, light, and sound in the Gardens of the Alcazar; the night visits to the Mosque-Cathedral of Cordoba; the walk through the history and culture of Cordoba; and the project "Route of the Patios of Cordoba" after its declaration as an intangible asset of the Humanity. These events encouraged and motivated tourists to spend the night, which gave them the opportunity to visit Madinat Al-Zahra.

In 2012 and 2013, there was a decrease in the number of tourists visiting the site, with it being least visited in the latter year. At the end of 2014, there was a decrease in the number of tourists to the site that changed the trend in 2015. From that year to the present, there has been a growing trend in the visits to the site, with an expected rebound given the recent registration by UNESCO.

All these tourist data show that the archaeological site of Madinat Al-Zahra, although it has been cared for and managed from the archaeological point of view, is not yet consolidated with respect to other monuments of the city with respect to tourist use and value.

*3.2. Hyphotesis*

The tourist satisfaction level is the result of different factors such as the tourist perception of the product and services they receive, and therefore their interest and motivation, or the expectations

generated before and during the trip (value and satisfaction received from the destination and of its main attractions).

To offer an attractive tourist destination it is necessary to have a deep knowledge of the tourist interests, the reasons that lead them to choose a particular destination, the value that is given to each attraction offered in the destination and the destination in general, as well as the degree of satisfaction with the products or services that the tourist receives, and with the destination in general. In this sense, the causal relationships between interest, motivation, value of the activities carried out in the destination, and tourist satisfaction have been widely analyzed.

Reference [48] offers a multitude of approaches to the creation of destination value, it presents a set of definitions, perspectives, and interpretations of how tourists create a value of their experience in tourism individually and collectively. Authors such as [49] or [50] have analyzed the relationship between interest and satisfaction degree regarding a tourist attraction, in the cases of Mauritania and Dubrovnik. In the study of [51], referring to dark tourism, it is concluded that an increase in curiosity motivation can result in a decrease in the restrictions of disinterest.

On the other hand, the results of [52] show that the emotional experiences of tourists act as antecedents of the general perception of the image and satisfaction evaluation. In this same work, the relationship between perceived satisfaction in the destination and the perceived value in the visit is also analyzed and it is concluded that the emotional experiences of tourists are a background to the general perception of the image and satisfaction evaluation.

Reference [53] studies the case of tourists from China and Russia and show that the destination value is positively related to the general image of a tourist destination and the intention to visit it. Also, the influence of the destination value on the value given to the heritage by the tourist has been analyzed [54]. The relationship between the perceived value and the satisfaction perceived by the tourist has also been analyzed by authors such as [55] for the case of the Korean heritage tourist.

Other works such as that of [56] show that the various attributes of the destination, and therefore, of its tourist attractions, influence positively or negatively the quality of the experience of Chinese tourists and their satisfaction.

Based on the previous literature review, the hypotheses tested that are shown in the diagram constructed in Figure 3 were formulated and are defined below, in which the associations between interest, motivation, value of the destination, and the Archaeological Ensemble are analyzed, as well as satisfaction for tourists who visit Cordoba and the new World Heritage Site.

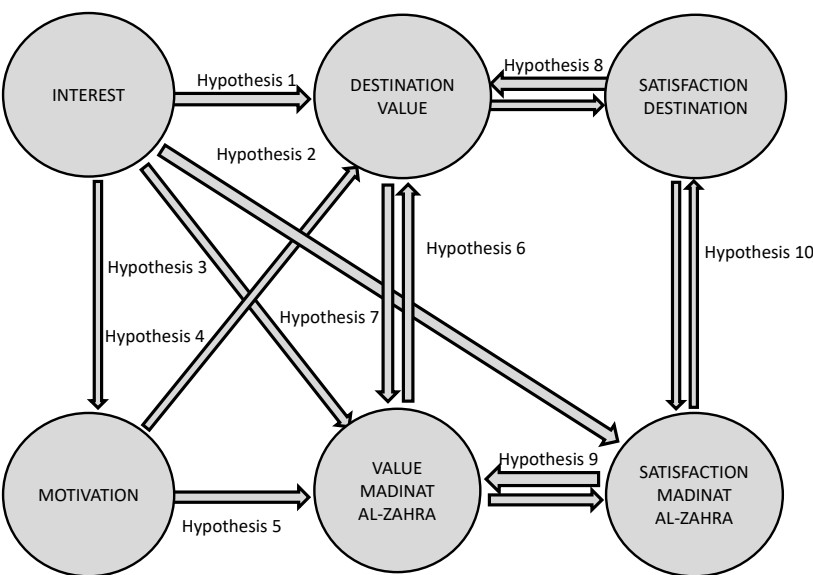

**Figure 3.** Research Model. Source: Authors, adapted from ACSI.

The general formulation of the hypothesis is:

$$\begin{cases} H_0 : \beta_{ij} = 0 \\ H_1 : \beta_{ij} \neq 0 \end{cases} \tag{1}$$

where, is the regression weight for a construct in the prediction of the other construct.

**Hypothesis 1:** *The interest of the tourist—does not influence (H10)—versus—influences significantly (H1a)—the destination value.*

**Hypothesis 2:** *The interest—does not influence (H20)—versus—influences, in a significant way (H2a)—the degree of satisfaction in the visit to Madinat al-Zahra.*

**Hypothesis 3:** *The interest shown by the tourist has not an influence (H30)—versus—has a significant influence (H3a)—on the perceived value of Madinat al-Zahra.*

**Hypothesis 4:** *The tourist motivation—does not influence (H40)—versus—influences, significantly, (H4a)—the destination value.*

**Hypothesis 5:** *The motivation—does not influence (H50)—versus—influences, in a significant way, (H5a)—the perceived value of Madinat al-Zahra.*

**Hypothesis 6:** *The value of Madinat al-Zahra of the tourist—does not influence (H60)—versus—influences significantly (H6a)—the destination value.*

**Hypothesis 7:** *The destination value of the tourist—does not influence (H70)—versus—influences significantly (H7a)—the value of Madinat al-Zahra.*

**Hypothesis 8:** *The satisfaction perceived in the tourist destination—does not influence (H80)—versus—influences, significantly, (H8a)—the perceived value of it.*

**Hypothesis 9:** *The perceived value in Madinat Al-Zahra tourist—does not influence (H90)—versus—influences, significantly, (H9a)—the satisfaction perceived in Madinat al-Zahra.*

**Hypothesis 10:** *The satisfaction perceived in Madinat Al-Zahra does not influence (H100)—versus—significantly influences (H10a)—the satisfaction perceived by the destination.*

*3.3. Materials and Methods*

The selected method for data collection was a structured questionnaire. In the first phase of the study, a pilot sample was selected to gather the necessary information for the fulfillment of the research objectives. A closed and self-administered questionnaire was chosen, among the available options for the collection of information. The items were formulated from previous studies [57,58] in order to guarantee the questionnaire validity. Once the elements were selected, they were evaluated by two groups of experts: three researchers in the tourism sector and four professionals from the tourism activity in the region. Therefore, the validity of the elements that form the constructs of the proposed model was verified through two ways. Once this was done, and based on the information from the pilot sample, the internal consistency of the questionnaire was analyzed.

The final survey was conducted in person, individually, and at different times during the morning, afternoon, and evening by random selection with national and foreign tourism quota, to tourists who visited the ruins of Madinat Al-Zahra and who had also visited other tourist sites in the city such as the Cathedral Mosque, the Synagogue, the Fortress of the Catholic Monarchs, the Courtyards Festival,

and knew the gastronomy of Cordoba. The survey team informed tourists about the purpose of the research, anonymity was assured, and they were asked to voluntarily participate in the study before starting the survey. Next, the tourists answered the questionnaire in their native language or in one that they spoke perfectly, from among the visitors who got off the bus that brought them back from their visit to the ruins of Madinat Al- Zahra, the only means of access to the monument.

The consistency of the final questionnaire was analyzed by calculating the Cronbach alpha coefficient for each construct, and for the total questionnaire, which in its final version consisted of 38 items grouped into 20 questions. The survey was conducted in the month of April 2018, through a non-probabilistic sampling by intentional quotas. The number of valid questionnaires was 375 in total, and with these a confidence level of 95.5% and a sampling error of 3.17% were obtained.

The questions of the first part of the questionnaire, socio-demographic profile and details of the trip, were closed. For the answers to the items of the second and third sections (used to measure interest, motivation, perceived value, and satisfaction), a Likert scale of 10 points was used, with 1 for the most negative response value and 10 for the most positive in some and from −5 to 5 for others (value and satisfaction) to offer the possibility of evaluating the item negatively. The value of global Cronbach's alpha coefficient calculated is 0.926, and for each of the constructs exceeds 0.75, which is therefore acceptable according to [59], who consider a scale to be acceptable if their Cronbach alpha is above 0.7 (Interest: 0.752, Motivation: 0.793, Destination value: 0.875, Satisfaction with destination: 0.8 and Medinat Al-Zahra value: 0.856).

In tourism research, variables such as motivation, interest, value of the tourist destination, and satisfaction of tourists are analyzed, variables that are not directly measurable, so other observable variables have to be used. Structural equation models allow researchers to estimate multiple dependency relationships and represent these relationships between unobserved or latent variables, considering the measurement error in the estimation process [60]. The methodology used for the construction of the indicators of the destination and the Madinat al-Zahra value, and the satisfaction in the destination and Madinat Al-Zahra is based on structural equations models. The data were tabulated and analyzed using the IBM SPSS 23 statistical software, and the estimates of the equations with IBM SPSS Amos 23. The model was designed (see Figure 3 above) based on the [61], and includes the observed and latent variables. Table 1 shows the latent variables, the observed variables, and the measurement errors of the proposed model. Table 2 defines each of the variables observed in the analyzed model.

**Table 1.** Non-observable variables used in the proposed model.

| | |
|---|---|
| Latent variables | Interest, Motivation, Destination value, Madinat Al-Zahra Value, Destination satisfaction and Satisfaction Madinat Al-Zahra. |
| Measurement errors | E1, E2, E3, E4, E5, E6, E7, E8, E13, E14, E15, E16, E17, E18, E20, E21, E22, E23, E24, E25, E26, E18, E19, E20, E21, E24, E25, E26, E28, E29, E30, E31, E32, E33, E34, E35, E36, E37, E38, E40, E41, E42, E43, E44, E45, E47, E49, E50, E51, E53 y E54. |

Source: Authors.

The final estimated model can be observed in Figure 4, in which the terms of disturbance have been included, which include the effects of the omitted variables, the measurement errors, and the randomness of the specified process. The regression coefficients, which represent the relationship between an exogenous and an endogenous latent variable, and between the endogenous latent variables among themselves, will be detailed in Table 3. The model was estimated using the generalized least squares method (GLS).

**Table 2.** Variables observed.

| Observed Variable | Definition |
| --- | --- |
| P7.1_visit Madinat al-Zahra | Visit Madinat al-Zahra. |
| P7.2_taste gastronomy | Taste the cuisine of the destination. |
| P7.3 know material heritage | Know the material heritage of the destination. |
| P7.4 know museums | Know museums. |
| P7.5_entertainment | Know the leisure and entertainment offer. |
| P7.6 relax | Relax. |
| P7.7 know city | Visit the city. |
| P7.8 visit friends and family members | Visit family and friends |
| P8.1 Madinat al-Zahra | Madinat al-Zahra. |
| P8.2 mosque | Cathedral mosque. |
| P8.4 hospitality treatment | Hospitality and treatment at the destination. |
| P8.5 information points | Tourist information points of the destination. |
| P8.6 signposting | Signposting of the monuments of the destination. |
| P8.7 accommodation | Accommodation used in the destination. |
| P8.8 gastronomy | Gastronomy of the destination. |
| P8.9 citizen security | Citizen security of the destination. |
| P8.10 cleanliness | Cleanliness of the destination. |
| P8.12 taxis | Public transport of the destination. |
| P8.13 cultural acts show | Cultural activities and shows of the destination. |
| P8.14 travel price | Prices in the destination. |
| P9.1 conservation | Degree of conservation of the architectural ensemble |
| P9.3 Time waiting bus ruins | Bus wait time to Madinat al-Zahra |
| P9.4 information received guided visit | Information received in the guided tour. |
| P9.5 Madinat al-Zahra shop | Souvenir store of the interior. |
| P9.6 explanatory video | Explanatory video of the site. |
| P9.7 accessibility | Accessibility. |
| P9.8 reception center visitors museum | Visitor reception center and museum. |
| P10 satisfaction | Visit to Madinat al-Zahra. |
| P11.1 cathedral mosque | Cathedral mosque. |
| P11.2 synagogue | Synagogue. |
| P11.3 fortress Christian monarchs | Fortress of the Christian monarchs. |
| P11.4 courtyards party | Courtyards party. |
| P12.1 cathedral mosque | Cathedral mosque. |
| P12.2 synagogue | Synagogue. |
| P12.3 fortress Christian monarchs | Fortress of the Christian monarchs. |
| P12.4 courtyards party | Courtyard party. |
| P12.5 gastronomy | Gastronomy. |
| P13 general satisfaction | Satisfaction with the visit to the architectural complex |

Source: Authors.

The literature recommends using multiple indicators to evaluate the fit of the model. Among the most used, we can highlight the ratio between the chi-square statistic and the degrees of freedom (CMIN / DF), the comparative adjustment index (CFI), the goodness of fit index (GFI), and the mean square error of approximation (RMSEA). The values of these goodness of fit statistics (CFI, GFI) generally vary between 0 and 1, with 1 indicating a perfect fit. Values higher than 0.9 suggest a satisfactory adjustment between the theoretical structures and empirical data, and values of 0.95 or higher, an optimal adjustment.

Finally, after considering the adjustment of the model, attention should be paid to the significance of the estimated parameters, analogous to the regression coefficients.

Based on the proposed model and once validated, the valuation indices of the latent variables (SIL)—perceived and expected quality of the destination and of Madinat al-Zahra, the perceived value of both, and the customer satisfaction—were calculated. The following arithmetic average, in which

each measurable variable is weighted through its coefficient with the latent variable, that is explained in the estimated model,

$$\overline{y_{ik}} = \frac{\sum\limits_{j=1}^{m} \lambda_{ij} X_{ijk}}{\sum\limits_{j=1}^{m} \lambda_{ij}} \qquad (2)$$

is the standardized coefficient between the latent variable, and the measurable variable. The value, calculated for each observation, is the estimated average value of each latent variable. The results obtained by this method have been scaled to express them on a scale standard from 0 to 100. This will allow estimation of a value for each latent variable, with which to quantify each one of them.

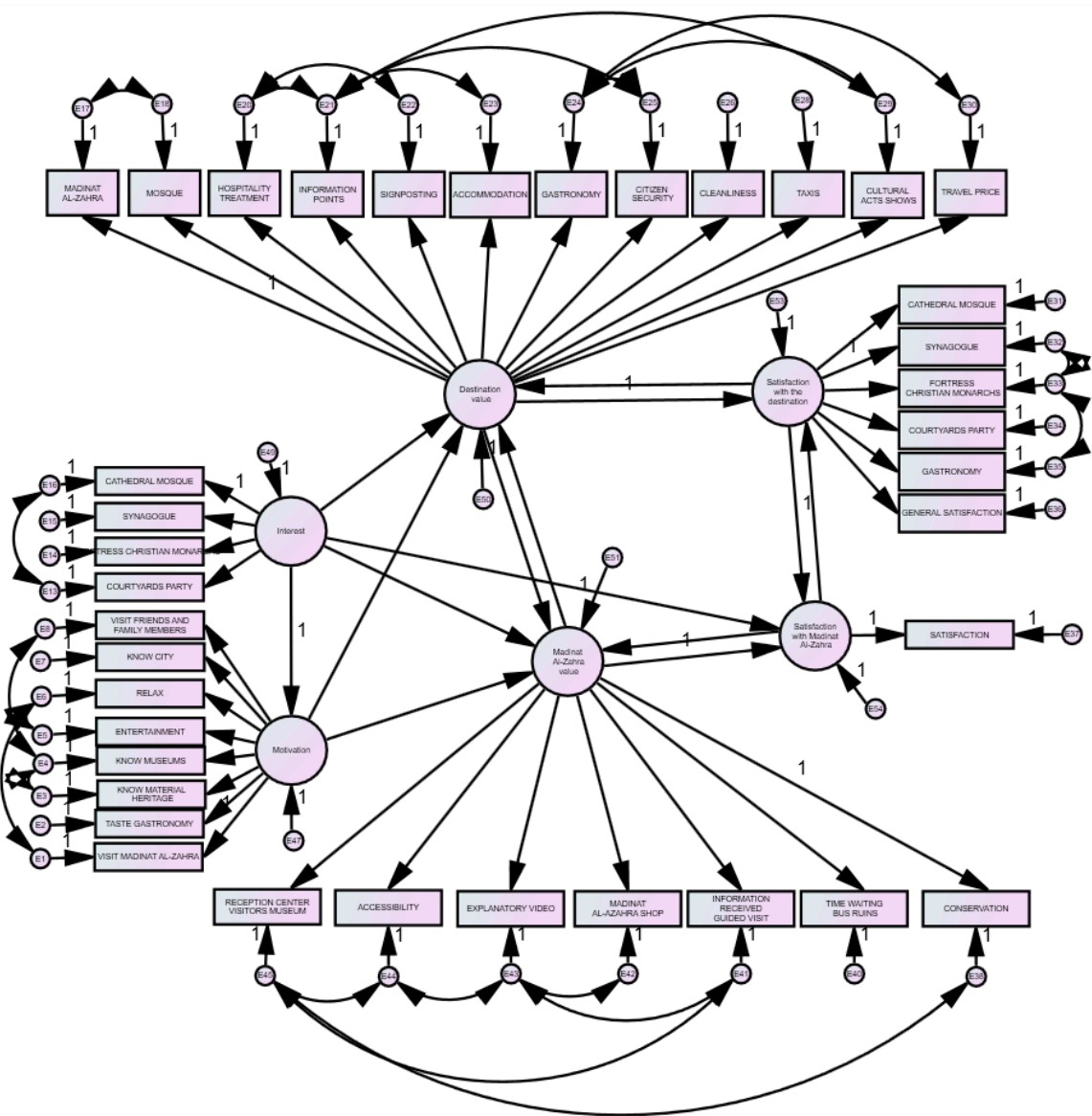

**Figure 4.** Path diagram of the proposed model. Source: Authors, adapted from the ACSI [61].

**Table 3.** Standardized structural coefficients of the variables observed on the latent.

| Latent Variable | Observed Variable | Standardized Coefficient |
|---|---|---|
| Interest | P11.1__mosque _cathedral | 0.750 |
| | P11.2_ synagogue | 0.584 |
| | P11.3_fortress_catholic_monarcs | 0.779 |
| | P11.4_courtyard_ party | 0.679 |
| Motivation | P7.1_visit_madinat_al_zahra | 0.622 |
| | P7.2_taste_gastronomy | 0.725 |
| | P7.3_ know_patrimony_material | 0.739 |
| | P7.4_know_museums | 0.689 |
| | P7.5_entertainment | 0.647 |
| | P7.6_relax | 0.624 |
| | P7.7_know_city_cordoba | 0.755 |
| | P7.8_visit_friends_family | 0.387 |
| Satisfaction destination | P12.1_mosque_cathedral | 0.681 |
| | P12.2_synagogue | 0.594 |
| | P12.3_fortress_catholic_monarcs | 0.746 |
| | P12.4_party_courtyard | 0.733 |
| | P12.5_gastronomy | 0.732 |
| | P13_general_satisfaction | 0.694 |
| Satisfaction Madinat al-Zahra | P10_satisfaction_madinat_al_zahra | 0.721 |
| Value Madinat al-Zahra | P9.1_conservation | 0.688 |
| | P9.3_time_waiting_bus_ruins | 0.738 |
| | P9.4_ information_rec_visit_guide | 0.736 |
| | P9.5_shop_madinat_al_zahra | 0.660 |
| | P9.6_explanatory video_site | 0.63 |
| | P9.7_accessibility_madinat_al_z | 0.638 |
| | P9.8_center recepti visit museum | 0.608 |
| Value_ Destination | P8.1_madinat_al_zahra | 0.608 |
| | P8.10_cleanliness | 0.701 |
| | P8.12_taxis | 0.733 |
| | P8.13_cultural acts shows | 0.703 |
| | P8.14_price_travel | 0.566 |
| | P8.2_mosque_ | 0.700 |
| | P8.4_hospitality _treatment | 0.671 |
| | P8.5_points_information_aten_vis | 0.729 |
| | P8.6_signporting | 0.712 |
| | P8.7_accomadtion | 0.711 |
| | P8.8_gastronomy | 0.691 |
| | P8.9_citizen_security | 0.690 |

Source: Authors.

*3.4. The Sample*

The descriptive analysis of the data collected leads us to the conclusion that most of the tourists who traveled in the spring of 2018 were young, since 85.3% were under 45 years old, with slightly more women (54.5%), and most of them have a higher education degree (53.4%). The distribution of the sample according to nationality was done intentionally, so that the distribution of domestic and foreign tourists was the same as that observed at the population level in 2017 (42.2% foreign tourists and 57.8% nationals), so that, of the sample formed by 354 valid questionnaires, 150 respondents were foreigners. Thirty-seven percent were visiting this city for the first time, and although most stayed in hotels and tourist apartments (55.1%) for an average of 4.8 days, it is surprising that just over 20% of tourists said they stayed at friends or relatives' house.

The main reasons for tourists to visit Cordoba, with an average of 8.22 and 8.06 out of 10 respectively, are knowing the city and its material heritage. So, the most valued aspects of the city by tourists are, with an average valuation higher than 8.5, the Cathedral Mosque and gastronomy, which are the tourist attractions, next to the Courtyard Party, which awaken more interest. On the contrary, the worst value is signposting, with an average score of less than 7.3.

In general, the valuation given to the monument under study, Madinat al-Zahra, is lower than that of Cordoba. The valuation given to the waiting time of the access buses with an average of 7.04 is especially improvable; while the most valued is the visitor reception center and the museum, with more than 8.5 points.

## 4. Results

For the proposed model, the relationship between the observed and latent variables is shown in Table 3. Also, the structural coefficients of the normalized model have been calculated.

The measurement of the indices of motivation. interest and perceived value will allow us to confirm the direct relationship between these constructs and the satisfaction of the tourist in the visit to the city of Cordoba.

Regarding the latent variables. in Table 4 and Figure 5 we can observe the intensity of measurement of the structural coefficients. as well as the limit probability that validates the significance of the relationship between the constructs of the proposed model.

**Table 4.** Standardized coefficients and p-values that support the hypotheses.

| Hypothesis | Latent Variable | Observed Variable | Standardized Coefficient | *p*-Value |
|---|---|---|---|---|
| Hypothesis 1 | INTEREST | VALUE_DESTINAT | 0.809 | 0.007 |
| Hypothesis 2 | INTEREST | SATISFACTION MADINAT_AL_ZAH | 0.458 | 0.037 |
| Hypothesis 3 | INTEREST | VALUE_MADINA_AL_ZAHRA | 0.474 | 0.000 |
| Hypothesis 4 | MOTIVATION | VALUE_DESTINAT | 1.521 | 0.011 |
| Hypothesis 5 | MOTIVATION | VALUE_MADINAT_AL_ZAHRA | −0.048 | 0.700 |
| Hypothesis 7 | VALUE_DESTINATION | VALUE_MADINATAL_ZAHRA | 0.808 | 0.000 |
| Hypothesis 6 | VALUE_MADINAT_AL_ZAHAR | VALUE_DESTINAT | −0.237 | 0.623 |
| Hypothesis 9 | VALUE_MADINAT_AL_ZAHAR | SATISFACTION_MADINAT_AL | 0.831 | 0.009 |
| Hypothesis 8 | SATISFACTION_DESTINATION | VALUE_DESTINAT | 1.499 | 0.050 |
| Hypothesis 10 | SATISFACTION_MADINAT_ AL | SATISFACTION DESTINATION | −0.171 | 0.079 |

Source: Authors.

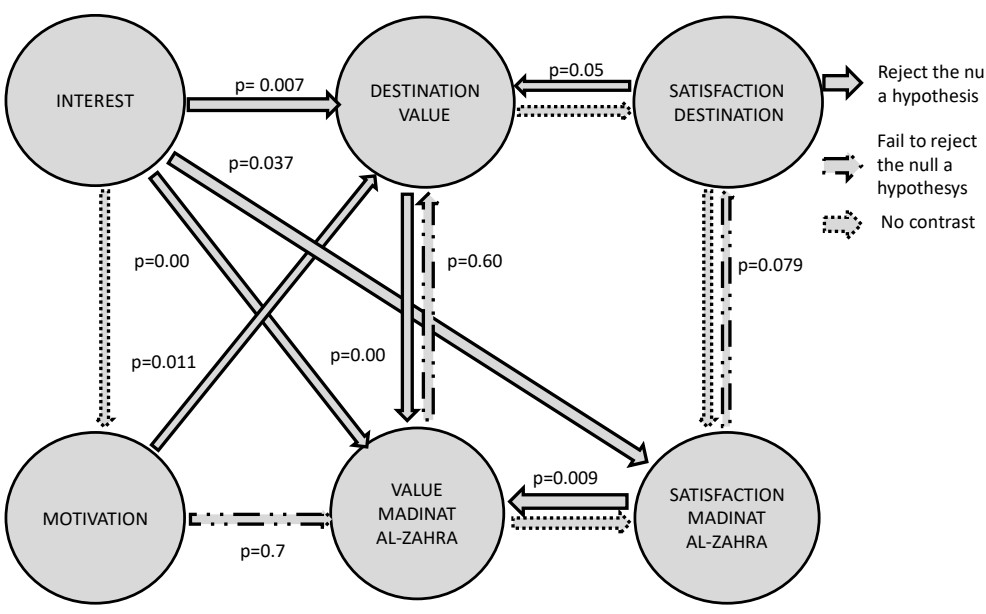

**Figure 5.** Path diagram of the proposed model with the p-value. Source: Authors.

To confirm the goodness of fit of the proposed model that supports the hypotheses formulated. the measures of absolute. incremental. and parsimony adjustment have been calculated (see Table 5).

**Table 5.** The goodness of fit of the model.

| Absolute Adjustment Measures | |
| --- | --- |
| Chi-square/DF | 1.779 |
| GFI | 0.832 |
| RMSEA | 0.047 |
| RMR | 0.416 |
| **Incremental adjustment measures** | |
| AGFI | 0.803 |
| **Measures of adjustment of parsimony** | |
| PNFI | 0.332 |
| PGFI | 0.711 |

Source: Authors.

Regarding the absolute measures that evaluate the global adjustments of the model [62]. the quotient between the Chi-Square coefficient and the degrees of freedom (1.799) is less than 2 and the RMSEA index is 0.047. both optimal. Some incremental adjustment measures such as the AGFI are sensitive to the number of indicators and are closely related to the GFI. with a value indicating an acceptable adjustment. With respect to the parsimony adjustment measures. PNFI and PGFI are significant above 0.06. also verified for the calculated model. On the basis of the results obtained. the model can be said to have an adequate goodness with respect to the adjustment indexes. which. together with the significance of the coefficients of the model. justifies its validity and applicability.

Once the model has been validated. and from the estimated coefficients of the model. the value indices of each of the latent variables have been estimated. obtaining a value for each individual surveyed. The means and standard deviations of the six latent variables are shown in Table 6.

**Table 6.** Mean and estimated standard deviations of the indices of the latent variables.

| Latent Variables | Average | Typical Deviation |
| --- | --- | --- |
| Interest | 82.650 | 11.9802 |
| Motivation | 72.633 | 14.9168 |
| Destination satisfaction | 83.455 | 12.2842 |
| Satisfaction of Madinat Al-Zahra | 76.186 | 16.1747 |
| Destination value | 78.500 | 13.1287 |
| Madinat Al-Zahra value | 77.923 | 14.0389 |

Source: Authors.

Obtaining this index allows us to compare the satisfaction and the assessment of the destination of visiting tourists to the city of Cordoba. as well as the satisfaction and appreciation of the destination of Madinat Al-Zahra with other World Heritage sites.

Particularly noteworthy is the interest (82.65%) and the satisfaction shown by tourists in the city of Cordoba (83.45%). However. the satisfaction shown by Madinat Al-Zahra. with 76.18%. is significantly lower than that of the city of destination. and also its value. with 77.92%. which means that the perceived value is lower than the expected value. This gap. although not very pronounced. can indicate the existence of expectations that are not covered by the visitors to the site. which should be considered.

On the other hand. satisfaction in the city of Cordoba. with 83.45% exceeds the destination value index (78.50%). so that the value perceived by tourists who visit it is greater than the expected value and. therefore. the level of satisfaction among the voting tourists of the city of Cordoba is high and exceeds their previous expectations. which may be an indication of a job well done by the institutions.

## 5. Discussion and Conclusions

The results of this study show the verification of most of the hypotheses. detailed below. which is based on the value of the estimated structural coefficients and the limit probabilities included in Table 4 that determine the results of the test at 5% level.

Hypothesis 1: The null hypothesis is rejected ($p = 0.007$). The direct influence that the tourist interest has on the perceived value of this towards the city of Cordoba is confirmed. A higher level of interest is expected to have a significantly positive effect towards the destination value perceived by tourists.

Hypotheses 2 and 3: The null hypotheses are rejected. Although with a higher probability in the case of satisfaction ($p = 0.037$). we can also affirm that the interest shown by tourists visiting the city of Cordoba positively influences both the perceived value and satisfaction experienced in the visit to the ruins of Madinat Al-Zahra. therefore a higher level of interest is expected to have a positive effect on the value and perceived satisfaction ($p < 0.001$).

Hypothesis 4: The null hypothesis is rejected ($p = 0.011$). The motivation that leads tourists to visit Cordoba positively and significantly influences the value perceived for the city.

Hypothesis 5: Failure to reject the null hypothesis ($p = 0.7$). It cannot be rejected that the motivation that leads tourists to visit Cordoba doesn't influence the value of the Archaeological Ensemble of Madinat Al-Zahra. This reinforces the existing belief among the authorities that the visitors of the monument are really motivated by the rest of the monuments of the city. but not so much by the site itself. which is visited as an additional activity.

Hypothesis 6: Failure to reject the null hypothesis ($p = 0.623$). In the same way. it is the case when analyzing the influence that the value of the site has on the value of the city. since it is not significant. This reinforces the perception that the monument is not integrated into the visitor mind as part of the city's heritage. because it is located outside the city and has a small weight in the value granted to this place.

Hypothesis 7: The null hypothesis is rejected ($p < 0.001$). From the contrast of this hypothesis. it can be deduced that the value perceived for the city of Cordoba has a positive influence. and with great intensity. on the value perceived for the Archaeological Complex. which can have a positive effect due to the high value granted by tourists to the city.

Hypothesis 8: The null hypothesis is rejected ($p = 0.05$). The satisfaction related to the city of Cordoba also influences its perceived value in a positive way. which is consistent. since when expectations about a destination are largely satisfied the perceived value increases. This is consistent with the results obtained by other works.

Hypothesis 9: The null hypothesis is rejected ($p = 0.009$). The value perceived for the Archaeological Ensemble of Madinat Al-Zahra also influences. in a significant way. the perceived satisfaction in the Archaeological Ensemble of Madinat Al-Zahra. A statistically significant structural coefficient confirms that the highest levels of value perceived in the ruins of the Archaeological Ensemble of Madinat Al-Zahra have a significant positive effect on the satisfaction perceived by tourists visiting the ruins of Madinat al-Zahra. which. again. coincides with that obtained in similar works.

Hypothesis 10: Failure to reject the null hypothesis ($p = 0.079$). Although it has been observed that it does not occur with the perceived value. there is an influence. even if not very significant. between the satisfaction perceived for the Archaeological Ensemble of Madinat Al-Zahra and the perceived satisfaction in the destination.

Work must be done. both by the public administration and private companies responsible for this World Heritage management. to improve the tourist experience. which increases the perceived value and. therefore. satisfaction. An improvement in the value perceived in Madinat Al-Zahra could lead to an increase in tourism expectations and then increase the additional tourist services (Hospitality. Hosting. Shop Purchase). through the perceived quality. which improves the satisfaction of the tourist even more [63–65].

A happy tourist with the visit made is a destination success. They start the trip with a previous photograph and some illusions about the destination. the final result can modify that illusion both positively and negatively. Consequently. the managing bodies. whether public or private. should improve information on the quality and image of tourist destinations.

Tourists who visit the city of Cordoba are mainly Spanish. under 45. they have visited the city before. and their main motivations are to know the city and. specifically. its heritage. With the inscription as World Heritage in July 2018 by UNESCO. an increase in interest to know it is expected. The objective should be that tourists that visit this monument reach levels of appreciation and satisfaction similar to the highest in the city. which boasts the Mosque and its gastronomy. It is necessary. therefore. to improve the signage of the city and the access to the Archaeological Ensemble of Madinat Al-Zahra. as well as from the visitor reception center. which enjoys. on the contrary. a great valuation.

From the results obtained in this research. it has been observed that the organisms in charge of the management of this new world heritage have a long way to go despite the recent nomination. Services. such as transportation. must be improved. as well as the communication and dissemination system of this invaluable site. in order to integrate it into the patrimonial complex of the city of Cordoba and overcome the isolation that currently exists. There is a great potential in the geographical and institutional environment in which Madinat Al-Zahra is located. since more than 80% of it is still undiscovered and it offers a great opportunity to become an essential objective in the visits of tourists in Cordoba.

Regarding the limitations of this study. other variables could have been included to explain satisfaction. such as climate. the effect of the advertising medium. prices. and emotional components. Future lines of research could focus on the intersection of information between supply and demand to provide data about an adequate balance in specific markets.

## 6. Concluding Summary

This study provides valuable instructions for local city managers and entrepreneurs. as well as for future public–private managers. which help to understand and measure the satisfaction of tourists through a model of structural equations and the obtained indexes. A new approach has been carried out to measure satisfaction and the perceived value based on a scale of zero to 100. these results can be very useful to compare different tourist places. The satisfaction obtained by the tourists who visit the city of Cordoba is high. However. the Archaeological Ensemble of Madinat Al-Zahra has to improve in aspects such as access to the site in order to get to levels of satisfaction of the set of tourist attractions that make up the city of Cordoba.

This research confirms that the motivation and interest. in a direct way; and the value in destination that is expected. indirectly. are important for a tourist destination that seeks visitor satisfaction. A tourist who experiences a high level of quality and satisfaction tends to recommend the visited place. These results are strategies and findings that any tourist place should consider in the planning and development of their products.

**Author Contributions:** A.H.-F. and R.H.-R.: Introduction, Theoretical Framework, Concluding Summary and Discussion and Conclusions; J.C.C.-R. and J.A.J.d.R.: Methodology, Results, Concluding Summary, Discussion and Conclusions.

**Funding:** This research received no external funding.

**Conflicts of Interest:** The authors declare no conflict of interest.

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
