# Peer review of "Tourist Motivations and Satisfaction in the Archaeological Ensemble of Madinat Al-Zahra"

_sustainability, doi:10.3390/su11051380_

Round 1

Reviewer 1 Report

The literature review should include more recent studies.

Reference no. 39 is not complete.

Author Response

Thanks for your comments. The review has been expanded with recent studies. Reference 39 has been completed (Now, 35).

Reviewer 2 Report

This paper clearly sets out the nature of the research project, its purpose and significance.  The history of the location and its importance are well presented, the language of the paper is clear and easy to understand. 

More detail on the selection process used in the identification of the candidates for interview. No information is presented on the informed consent process.

Author Response

The explanation of the followed selection process has been expanded: lines 327 - 336.

Reviewer 3 Report

The organisation is poor, important to review the overall organisation in order to be more understandable the underlying reasoning

The method should be explained in more detail

Serious gaps in the literature review

lack of a clear link with the concept of sustainability

the abstract has to be reviewed, possibly avoiding achronyms 

Author Response

The article has been completely restructured in order to make it more understandable, the contrast hypotheses analyzed and the methodology have been justified and detailed. The  literature review has been expanded. The acronyms of the summary have been removed.

Reviewer 4 Report

The article deals with a subject of great interest from a tourist perspective such as archaeology. However, it is a confusing document, badly organized and lacks the theoretical basis to support the different hypotheses it puts forward. It follows an unconventional exhibition structure in the scientific articles, which contributes to making it more difficult to follow and understand the study carried out.

In the first place, the introduction should begin by addressing the importance of archaeology in tourism and citing different previous research to support the scientific interest of the work. Subsequently, a generic reference can be made to the destination on which the study is to be carried out, which should be developed in greater depth in the section on methodology. In the introduction some generic objectives are proposed which I consider are not sufficiently clear and precise and, in some of them, they are not dealt with in the rest of the study nor can they be the object of the same (e.g. lines 46-47, provide recommendations to public and private entities).

There is also a descriptive study with another of structural equations, which means that the fundamental part of the work is lost.

The theoretical framework is presented on page 5 in a generic way and, subsequently, tests and hypotheses are presented in a confusing way and without theoretical support to support them. I consider that too many tests are posed with hypotheses posed in an inadequate and confusing way. It is preferable to pose the hypotheses one by one as is usually done in scientific research, with the corresponding theoretical support. This means that the theoretical model proposed is not clear, and direct and inverse relations are apparently proposed at the convenience of the constructs. For example, why is the relationship between the interest and satisfaction of Medinat Al-Zahra analysed and not with respect to the satisfaction of destiny? How are two-way tests carried out between the value of destinies and satisfaction? What is the difference between the construct value and the construct satisfaction? How can partial destinies be linked within more global destinations such as the case of Cordoba as a whole? Another aspect that should be dealt with beforehand is to define the interest and motivation of tourists about a destination as well as to determine how both concepts are measured.

Another big problem of this article is that the scales of variables used to measure each construct are not clearly defined. Table 2 is presented where the variables and definitions for each construct are not clear. In addition, the variables are defined as dependent, when they are really independent variables in their great majority. It is not clear which are dependent and which are independent. However, in structural analysis it is preferable to talk about constructs and variables that are used to measure each of them. Then, depending on the statistical analysis to be applied in function of the hypothesized relationships, they may be explanatory variables or variables to be explained.

Since the variables are not clearly defined, the complex path analysis is much more difficult to evaluate and understand. In relation to the results, no analysis is made of the reliability and validity of the different scales of variables used, except in the case of Cronbach's Alpha. The analysis of results focuses on the tests that have been proposed but which should actually be hypotheses. The explanations given for accepted or rejected hypotheses are very specific and do not derive theoretical or practical implications. Furthermore, they are not related to previous studies to check whether the same or different results are obtained. Point 6 is introduced on descriptive analysis, which brings more confusion to the article, because of its content and because it is preferable to focus on structural analysis. In short, there is no adequate discussion of the results.  Furthermore, due to the fact that the methodology and variable scales are not clearly defined, it is not possible to evaluate whether the results are adequate, whether the method is suitable and whether the constructs proposed are adequately measured.

It is a pity that a work with acceptable background work cannot be adequately presented in a scientific article because it has not been properly specified, structured and defined. Sometimes less is more.

Author Response

The article has been completely restructured according to the recommendations and the hypothesis of contrast has been justified through a literature review. The unaddressed objectives proposed in the introduction have been removed (lines 46-47 of the previous document). The hypothesis of contrast are proposed jointly since the model of structural equations is built globally on them. The model used is based on the American Customer Satisfaction Index [58]. The relationship between the value and the satisfaction of the destination and a part of it have been evaluated before (Jamaludin et al., 2018; Moon et al, 2018; Su, 2019).

Some relationships not included in the model can be analyzed indirectly, for example: the relationship between interest and satisfaction at the destination can be assessed through the destination value. The number of degrees of freedom available in the creation of the model does not allow the inclusion of all the associations. Therefore, the direct associations that were wanted to be evaluated in the hypotheses have been included.

The explanation of the nature of the variables has been extended (lines 342-346). The Dependent-Independent notations have been removed and replaced by constructs and observable variables. The validation process of the instrument has been detailed (lines 318-326, 337-341 and 346-350).

Round 2

Reviewer 3 Report

The approach to the subject is confused and confusing. The knowledge related to heritage tourism, archaeological tourism etc which one considers to be reflected in the literature review, is very poor. The application of the methods appears out of the overall narrative, and it is not justified enough.

Author Response

Thank you very much

Reviewer 4 Report

I congratulate the authors on their good work.

Author Response

The English edition has been revised and improved. The bibliographical references 27-28-38 have been included in the theoretical framework and in the methodology.